# The clonal structure and dynamics of the human T cell response to an organic chemical hapten

Tahel Ronel[1,2], Matthew Harries[3,4], Kate Wicks[3], Theres Oakes[1], Helen Singleton[3], Rebecca Dearman[3], Gavin Maxwell[5], Benny Chain[1,6]*

[1]Division of Infection and Immunity, University College London, London, United Kingdom; [2]Cancer Institute, University College London, London, United Kingdom; [3]Faculty of Biology, Medicine and Health, University of Manchester, Manchester, United Kingdom; [4]Salford Royal NHS Foundation Trust (Dermatology Centre), Salford, United Kingdom; [5]Safety and Environmental Assurance Centre, Unilever, Colworth Science Park, Bedford, United Kingdom; [6]Department of Computer Science, University College London, London, United Kingdom

**Abstract** Diphenylcyclopropenone (DPC) is an organic chemical hapten which induces allergic contact dermatitis and is used in the treatment of warts, melanoma, and alopecia areata. This therapeutic setting therefore provided an opportunity to study T cell receptor (TCR) repertoire changes in response to hapten sensitization in humans. Repeated exposure to DPC induced highly dynamic transient expansions of a polyclonal diverse T cell population. The number of TCRs expanded early after sensitization varies between individuals and predicts the magnitude of the allergic reaction. The expanded TCRs show preferential TCR V and J gene usage and consist of clusters of TCRs with similar sequences, two characteristic features of antigen-driven responses. The expanded TCRs share subtle sequence motifs that can be captured using a dynamic Bayesian network. These observations suggest the response to DPC is mediated by a polyclonal population of T cells recognizing a small number of dominant antigens.

*For correspondence:
b.chain@ucl.ac.uk

## Introduction

Diphenylcyclopropenone (DPC) is an example of a hapten, a small organic compound that reacts with biological macromolecules including proteins to form immunogenic conjugates. It is a potent skin sensitizer (*Mose et al., 2017a*; *Stute et al., 1981*) and, since it is non-mutagenic in the AMES assay (*Wilkerson et al., 1987*), it has been used as an immunostimulant in the treatment of warts (*Buckley et al., 1999*), melanoma (*Read et al., 2017*), and alopecia (*Ashworth et al., 1989*; *Karanovic et al., 2018*; *Lee et al., 2018*). DPC can also act as an adjuvant to conventional immunization (*von Moos et al., 2012*). Because of its clinical applications, and because it is not commonly found in the environment, DPC serves as an interesting model to study primary and secondary responses to chemical haptens in humans, although the confounding effects of any underlying clinical condition in the treated individuals must obviously be considered. Clinically, repeated exposure results in rapid sensitization, but the magnitude of the resulting allergic contact dermatitis (ACD) does not continue to increase during repeated exposure (*Mose et al., 2017a*), suggesting the existence of regulatory processes. Global expression profiling at the site of elicitation suggested a predominantly Th1/Th17 type T cell infiltration, whose dynamics mirrored the clinical and histopathological picture (*Mose et al., 2017b*). High serum interleukin-4 (IL4) and low interleukin-12 (IL12) were observed in patients with alopecia showing a favorable response to DPC treatment.

As part of an extended project to understand more fully the immunological events leading to ACD, and hence generate better tools for chemical risk assessment (*Kimber et al., 2012*), we have measured the T cell receptor repertoire (TCRrep) changes in ex vivo blood samples collected from patients before and after therapeutic sensitization with DPC. In a previous study, we used a newly developed quantitative, robust experimental and computational pipeline (*Oakes et al., 2017a*) to measure changes in the TCRrep following in vitro re-stimulation of blood from individuals allergic to the environmental contact allergen paraphenylene diamine (*Oakes et al., 2017b*). A limitation of the study was that the extent of allergen exposure prior to challenge was unknown. Here we extend these studies to document in vivo changes to the TCRrep during primary and secondary sensitization with DPC in a small cohort of patients with alopecia. Exposure to DPC results in transient expansions of a set of related TCR sequences in the peripheral blood. The number of responding TCRs correlates with the extent of sensitization and is not observed in unexposed individuals. The results suggest that exposure to a contact allergen in vivo stimulates an initial polyclonal expansion of antigen-specific responding T cells which, however, does not increase on further repeated exposure to antigen.

## Results

We carried out TCRseq on unfractionated blood samples from 29 patients with alopecia (*Table 1*).

Samples were collected before (referred to as the pre-sensitization [PS] sample, week 0, n = 25 samples) and at one to three time points after sensitization with DPC (*Figure 1A*), taken at 2 weeks (PT1, n = 23), 6 weeks (PT2, n = 18), and approximately 24 weeks (PT3, n = 17) post-sensitization. As controls, we carried out TCRseq on five healthy volunteers, for which we had time points that matched the 0-, 2-, and 6-week time points of the DPC sensitized group. We obtained a total of 24,431,855 distinct TCR sequences, of which 17,451,853 were unique across all samples. The number of TCR sequences obtained varied between samples (mainly due to small differences in volume of individual libraries when preparing the pooled library for sequencing) and are shown in *Supplementary file 1*. The median number of TCRs per sample was 95,638 (range 23,224–690,600) and the median number of unique TCRs per sample was 75,111 (range 19,095–342,336). Analyses included every patient for whom the relevant time points (and patch test data where relevant) were available, and this number is indicated in the respective figure legends.

### Repeated exposure to DPC does not alter the global structure of the peripheral blood TCR repertoire

The major T cell sub-populations (CD4/CD8, naive, central memory, effector memory, and effector memory RA revertant) (*Fletcher et al., 2005*) were quantified by flow cytometry and did not change significantly after exposure to DPC (*Figure 1B*, two-sided paired t-tests corrected for multiple testing, p=0.48, p=0.74, p=0.49, and p=0.74 for CD4, and *Figure 1C*, p=0.11, p=0.72, p=0.72, and p=0.37 for CD8 naive, central, effector, and effector memory RA revertant cells respectively). Similarly, TCR repertoire diversity, captured by the Shannon diversity index, or the clonal expansion captured by the Gini inequality coefficient, did not differ significantly before and after exposure to DPC (*Figure 1D,E*, Kruskal–Wallis rank sum tests, p=0.89 and p=0.90 respectively), nor between the different time points post-sensitization (*Figure 1G,H*, Kruskal–Wallis rank sum tests, p=0.96 and p=0.95 respectively). Finally, the number of TCRs found at frequencies above 1 in 1000, which correspond to the most abundant 6% of the repertoire on average, did not change as a result of sensitization (*Figure 1F,I*, Kruskal–Wallis rank sum tests, p=0.12 and p=0.91 respectively). The corresponding analyses for the alpha chain sequences are shown in *Supplementary file 2*. There was therefore no evidence that exposure to DPC, a potent skin sensitizer, caused global alterations in the structure of the TCR repertoire.

### Sensitization with DPC induces a transient expansion in the frequency of a small subset of the TCR repertoire

We next looked for evidence of changes in individual TCR sequences following exposure to DPC. We plotted the abundance of each TCR before sensitization and after sensitization (at PT1) (representative examples are in *Figure 2A*; all individuals in *Supplementary file 3*). We observed that in a number of individuals, there was a population of TCRs which were absent before sensitization and

**Table 1.** Demographics of the study population.

| ID | Age | Sex[1] | Alopecia[2] | TCRseq sets[3] | Flow cytometry[4] | Patch test scores[5] |
|----|-----|-----|-----------|--------------|-----------------|--------------------|
| 1 | 50–59 | F | AA (>50%) | 0 | | X |
| 2 | 10–19 | M | AU | 0 | | X |
| 3 | 20–29 | F | AT | 3 | | X |
| 4 | 50–59 | F | AT | 4 | X | X |
| 5 | 50–59 | M | AA (>50%) | 3 | X | X |
| 6 | 40–49 | M | AA (>50) | 4 | X | X |
| 7 | 40–49 | M | AU | 4 | X | X |
| 8 | 30–39 | F | AA (>50%) | 3 | | X |
| 9 | 30–39 | F | AT | 0 | X | X |
| 10 | 20–29 | F | AT | 1 | X | |
| 11 | 30–39 | M | AU | 1 | | |
| 12 | 20–29 | F | AA (>50%) | 0 | X | X |
| 13 | 30–39 | F | AT | 1 | X | |
| 14 | 30–39 | F | AA (<50%) | 3 | | X |
| 15 | 40–49 | F | AU | 4 | X | X |
| 16 | 40–49 | F | AA (<50%) | 4 | X | X |
| 17 | 50–59 | F | AT | 3 | | X |
| 18 | 20–29 | F | AT | 1 | | |
| 19 | 50–59 | F | AU | 2 | | |
| 20 | 50–59 | F | AU | 4 | | X |
| 21 | 10–19 | M | AA (>50%) | 3 | | X |
| 22 | 30–39 | F | AA (>50%) | 4 | | X |
| 23 | 50–59 | F | AU | 4 | | X |
| 24 | 40–49 | F | AA (<50%) | 4 | | X |
| 25 | 40–49 | F | AT | 3 | | X |
| 26 | 60–69 | F | AA (<50%) | 2 | | |
| 27 | 30–39 | F | AA (>50%) | 1 | | |
| 28 | 30–39 | M | AA (<50%) | 3 | | X |
| 29 | 50–59 | F | AT | 4 | | X |
| 30 | 60–69 | F | AT | 3 | | X |
| 31 | 30–39 | F | AA (>50%) | 1 | | |
| 32 | 20–29 | F | AA (<50%) | 0 | | X |
| 33 | 40–49 | F | AU | 3 | | |
| 34 | 40–49 | F | AA (>50%) | 3 | | |

[1]F: female; M: male; [2]AA: alopecia areata (<50% or >50% scalp involvement); AT: alopecia totalis; AU: alopecia universalis. [3]The number of time points for which TCRseq data was obtained. [4]The patients for whom PBMC flow cytometry data were available. [5]The patients for whom patch test scores were available.

present at relatively high abundance after sensitization (indicated by the pink dots in the left panels of *Figure 2A*). This change is seen clearly in the abundance profile of all TCRs that were absent in the pre-sensitization sample, but present after sensitization (*Figure 2B*). On the basis of these profiles, we counted the proportion of TCRs absent at PS and present at PT1 with an abundance of eight times or above. The remainder of this study focuses on this population, which we refer to as PT1 expanded. The percentages of PT1 TCRs which are expanded in each individual are summarized for patients and healthy volunteers in *Figure 2C*. For comparison, we calculated the percentage of TCRs absent at PT1 but present at PS with an abundance of eight times or above (referred to as PS

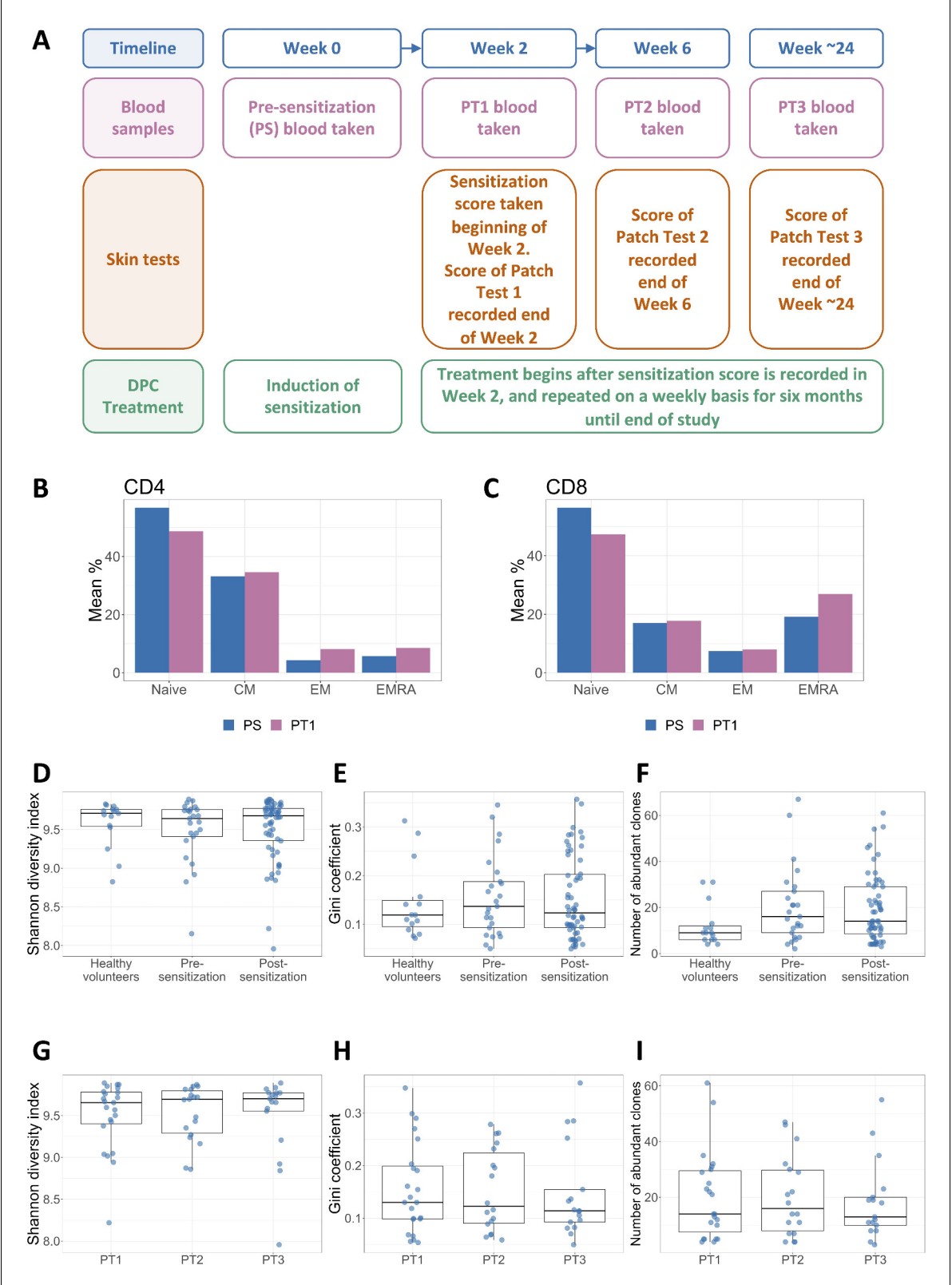

**Figure 1.** Repeated exposure to diphenylcyclopropenone (DPC) does not alter the global structure of the peripheral blood T cell receptor (TCR) repertoire. (A) The study outline showing when bloods were drawn, skin tests were performed, and DPC treatment was applied during the ~24 weeks of participation in the study. The four time points at which sensitization scores were recorded and blood samples were taken are referred to throughout the paper as PS (pre-sensitization, week 0), PT1 (Patch Test 1, week 2), PT2 (Patch Test 2, week 6) and PT3 (Patch Test 3, around week 24). In all cases

*Figure 1 continued on next page*

*Figure 1 continued*

bloods were drawn prior to application of the patch test. (B) and (C) The PS and PT1 blood samples of 10 patients were analyzed using flow cytometry. The mean percentage of total naive, central memory (CM), effector memory (EM), and effector memory RA (EMRA) expressing cells in the (B) CD4 and (C) CD8 compartments are shown. Paired t-tests with Benjamini–Hochberg correction for multiple testing were performed to check for significant differences between the pre- and post-sensitization cell number distributions for each subpopulation. All p-values were considerably higher than the 0.05 significance threshold. (D) The Shannon diversity index of the healthy volunteers (n = 15 samples from five individuals), pre-sensitization (n = 25), and post-sensitization (n = 58; from all three time points) TCR repertoire samples. All samples were randomly subsampled to the minimum sample size (21,838 beta TCRs), and the Shannon diversity index of the subsample was then calculated. Each sample is represented by a dot. The box plots show the median, and lower and upper quartiles of each group. Differences in the distribution of the three groups were tested using a Kruskal–Wallis rank sum test and were non-significant (p=0.87). (E) The Gini inequality coefficient of the healthy volunteers, pre-sensitization and post-sensitization TCR repertoire samples, subsampled as in (D). Differences in the distribution of the three groups were tested using a Kruskal–Wallis rank sum test and were non-significant (p=0.89). (F) The number of TCRs that appear with a frequency of 1/1000 or higher in each sample (termed 'abundant TCRs'), for the healthy volunteers, pre-sensitization and post-sensitization samples, subsampled as in (D) and (E). A Kruskal–Wallis rank sum test revealed no statistical difference between the groups (p=0.14). (G)– (I) Sensitized samples were separated according to time point: PT1 (n = 23), PT2 (n = 18), and PT3 (n = 17). The Shannon diversity index (G), the Gini coefficient (H), and the number of abundant clones (I) of these subsamples were then calculated. Kruskal–Wallis rank sum tests were used to compare between the three groups in each case. All tests showed no statistically significant difference, with p-values p=0.97, p=0.96, and p=0.90 respectively.

expanded). The range of percentages of expanded TCRs at PT1 is clearly much greater in the patients than in the healthy volunteers (between 0 and 2 weeks), although the medians are not significantly different (p=0.16 and p=0.47 for alpha and beta respectively, Mann–Whitney). The percentage of expanded TCRs at PT1 in the patients is significantly larger than the percentage of TCRs which were expanded at PS and absent at PT1 (p=0.02 for alpha, p=0.03 for beta, Wilcoxon signed–rank). This is not the case for the healthy volunteers (p=0.79 for alpha, p=0.59 for beta, Wilcoxon signed–rank). Similar results were obtained setting the expansion threshold at 16 or 32. The number of PT1 expanded TCR alpha and TCR beta sequences was highly correlated (Spearman's rho = 0.85, p<0.0001, *Supplementary file 3*).

After the initial increase post-sensitization, the number of expanded TCRs remained rather constant over the later time points. There was no evidence that the total number of expanded TCRs increased with time, despite the fact that individuals were exposed to repeated therapeutic stimulation with sensitizer on a weekly basis for the period of the study in addition to the patch test applications (*Figure 1A*) (Kruskal–Wallis rank sum test, p=0.92 alpha and p=0.36 beta). However, this apparent overall stability hid dramatic dynamic changes in the frequency of individual TCRs within individual patients. The majority of PT1 (2 weeks) expanded TCRs decreased or returned to baseline at subsequent time points (*Figure 3A*). The majority of PT2 expanded TCRs present at 6 weeks were not expanded at 2 weeks, peaked at 6 weeks, and decreased or returned to baseline by the time the PT3 sample was taken at around 24 weeks (*Figure 3B*). These dynamics are shown for the beta chain sequences of 10 patients for whom all four time points were available (the corresponding analysis for five healthy volunteers is in *Figure 3C*). The alpha chain sequences of the same patients behaved similarly (*Supplementary file 4*). The exposure to repeated doses of DPC therefore induced large, but predominantly transient, changes in the frequencies of a small proportion of the TCRs.

## TCR expansion after exposure to DPC correlates with the magnitude of skin sensitization

We hypothesized that the expanded TCRs identified above might be functionally related to development of ACD. Exposure to DPC induced skin sensitization (patch test scores of + or higher) in almost all individuals but the magnitude of the response varied significantly between individuals (*Figure 4A*). Interestingly, the maximum response was usually observed at the first patch test, and declined or remained constant thereafter despite repeated exposure (*Figure 4B*). The sensitization reaction, therefore, like the number of expanded TCRs, did not continue to increase despite repeated exposure to antigen. In consequence, the magnitude of the response was very diverse at the beginning (+++, ++, or +) but converged toward a score of + or below by week 24.

We plotted the number of PT1 expanded TCRs (abundance ≥8) in individuals with varying strengths of sensitization as a function of the initial skin response recorded at the site of application (*Figure 4C*, left panels) or at the first patch test (*Figure 4C*, center panels). The number of PT1

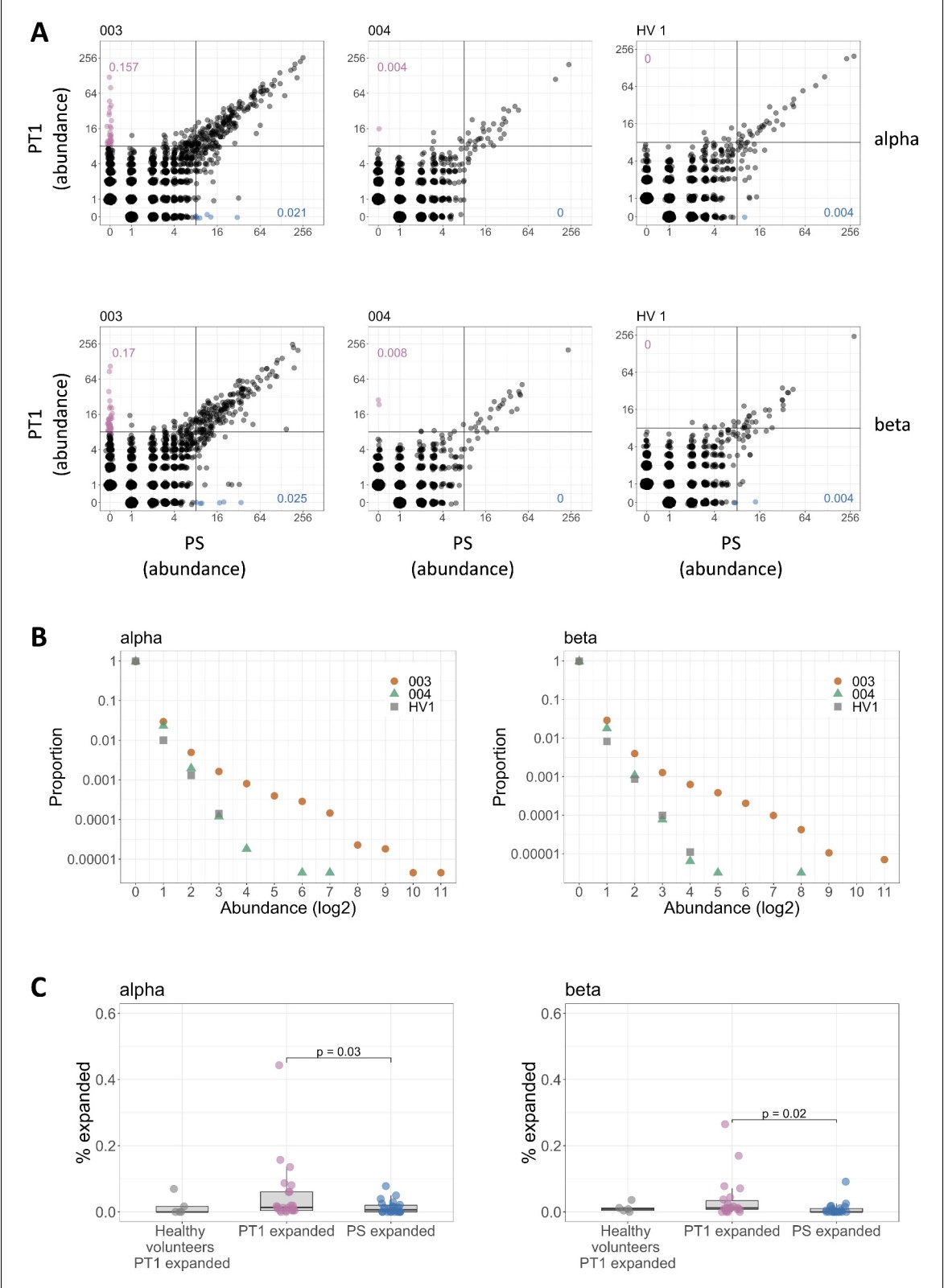

**Figure 2.** Sensitization with diphenylcyclopropenone (DPC) induces a transient expansion in the frequency of a small subset of the T cell receptor (TCR) repertoire. (**A**) The abundance distribution of TCRs at PS and PT1. All samples were subsampled to the same number of TCRs (28,000). Each unique TCR is represented by a dot, and the axes represent the number of times it is observed in the PS (x-axis) and PT1 (y-axis) sample of the same individual. The left and center panels represent two representative individuals. The right panels show one representative healthy volunteer at 0 and 2 weeks

*Figure 2 continued on next page*

*Figure 2 continued*

(equivalent timings to PS and PT1). The plots for all individuals are shown in **Supplementary file 3**. The pink dots identify a population of TCRs absent in the PS sample and expanded (abundance ≥8) in the PT1 sample. The blue dots identify a population of TCRs absent in the PT1 sample and expanded (≥8) in the PS sample. The numbers indicate the percentage of PT1 expanded TCRs (pink) and PS expanded TCRs (blue). (**B**) The abundance distribution profile of TCRs present at PT1 and absent at PS corresponding to the left panel in (**A**) (orange), middle panel (green), and right panel (gray). The y-axis shows the proportion of the TCRs which are found at the abundance indicated by the x-axis. (**C**) The distribution of the percentages of PT1 expanded (≥8) TCRs in healthy volunteers (n = 5) and in sensitized individuals (n = 22), and the distribution of the percentages of PS expanded (≥8) TCRs in sensitized individuals (n = 22), all subsampled as in (**A**). Bars show Wilcoxon signed–rank comparisons.

expanded TCRs was positively correlated to the patch test score. The correlation was positive but did not reach significance at an expansion threshold of 4, but remained significant when using a threshold abundance of 16 or 32. In contrast, there was no significant association between the number of PS expanded TCRs and the PT1 patch test score (**Figure 4C**, right panels). Although we measured the proportion of expanded TCRs, and not the absolute number, it remained possible that the total number of TCRs in each sample could affect the percentage of expanded TCRs. However, there was no significant correlation between the proportion of PT1 expanded TCRs (abundance ≥8) and the total number of TCRs sequenced in each sample (Spearman's correlation rho = 0.18 [p=0.4] and rho = 0.22 [p=0.3] for alpha and beta respectively). We also carried out repeated (10 times) subsampling of each sample to the size of the smallest sample. We observed a mean positive correlation (Spearman's rho) between the patch test scores and the percentage of PT1 expanded TCRs in the subsamples at a threshold of 8 (0.4 [alpha] and 0.3 [beta]), but the p-value did not reach significance (p=0.18 and 0.22 respectively). At a threshold of 16, the correlations were 0.54 (p=0.04) and 0.32 (p=0.2), and at a threshold of 32, the correlations were 0.58 (p=0.01) and 0.61 (p=0.006) for alpha and beta respectively. Overall, the qualitative pattern of greater number of PT1 expanded TCRs remained the same after subsampling although the smaller data sets did alter the magnitude of the correlations and the threshold at which they became significant.

The correlation between the number of expanded TCRs at all time points (in each case measured as change relative to pre-sensitization frequency) and the sensitization/patch test scores at all time points is summarized in **Figure 4D**. As illustrated above, the numbers of expanded TCRs at PT1 were correlated with the strength of the reaction recorded at the sensitization site, and with the patch test score at PT1 (2 weeks), as well as PT2 (6 weeks). The correlation was lost at PT3 (24 weeks), perhaps reflecting the greatly decreased range of patch test scores at this time point (**Figure 4B**).

## The expanded TCRs show characteristics of antigen-driven responses

We hypothesized that the population of TCRs that are found at increased frequency post-exposure to DPC may be enriched for T cells which bind to DPC, or DPC-modified peptides. Antigen-specific sets of TCRs frequently share sequence features, including skewed use of V and J regions, and similarities in CDR3 sequence (**Dash et al., 2017**; **Davis et al., 1995**; **Glanville et al., 2017**; **Pogorelyy et al., 2019**; **Sun et al., 2017**; **Thomas et al., 2014**). The relative V and J gene usage profiles within the expanded set of TCRs compared to the pre-sensitization repertoire for that individual are shown for each patient in **Figure 5**. We used a non-parametric bootstrapping approach to determine which V and J genes were statistically significantly skewed in the repertoire of the expanded TCRs. We compared the observed proportion of each V and J gene in the PT1 expanded TCRs (≥8) with 1000 random sets of the same number of TCRs sampled from the pre-sensitization repertoires of the same individuals. V and J genes that ranked in the top 50 out of 1001 were considered significantly under-represented at the 0.05 significance level, and genes in the bottom 50 ranks as significantly over-represented. Several examples of skewed V and J gene usage were observed, both in TCR alpha and TCR beta (pink dots). Due to the relatively small number of expanded TCRs in some patients, under-represented genes were very common as a result of sampling, and hence less statistically robust. We therefore show only over-represented V and J genes. In addition, genes that were over-represented by a similar analysis on the set of all expanded TCRs (generated by combining the individual patient expanded sets and taking unique TCRs) are also shown (**Figure 5**, pink gene names). Interestingly, some V and J regions were skewed in several different individuals,

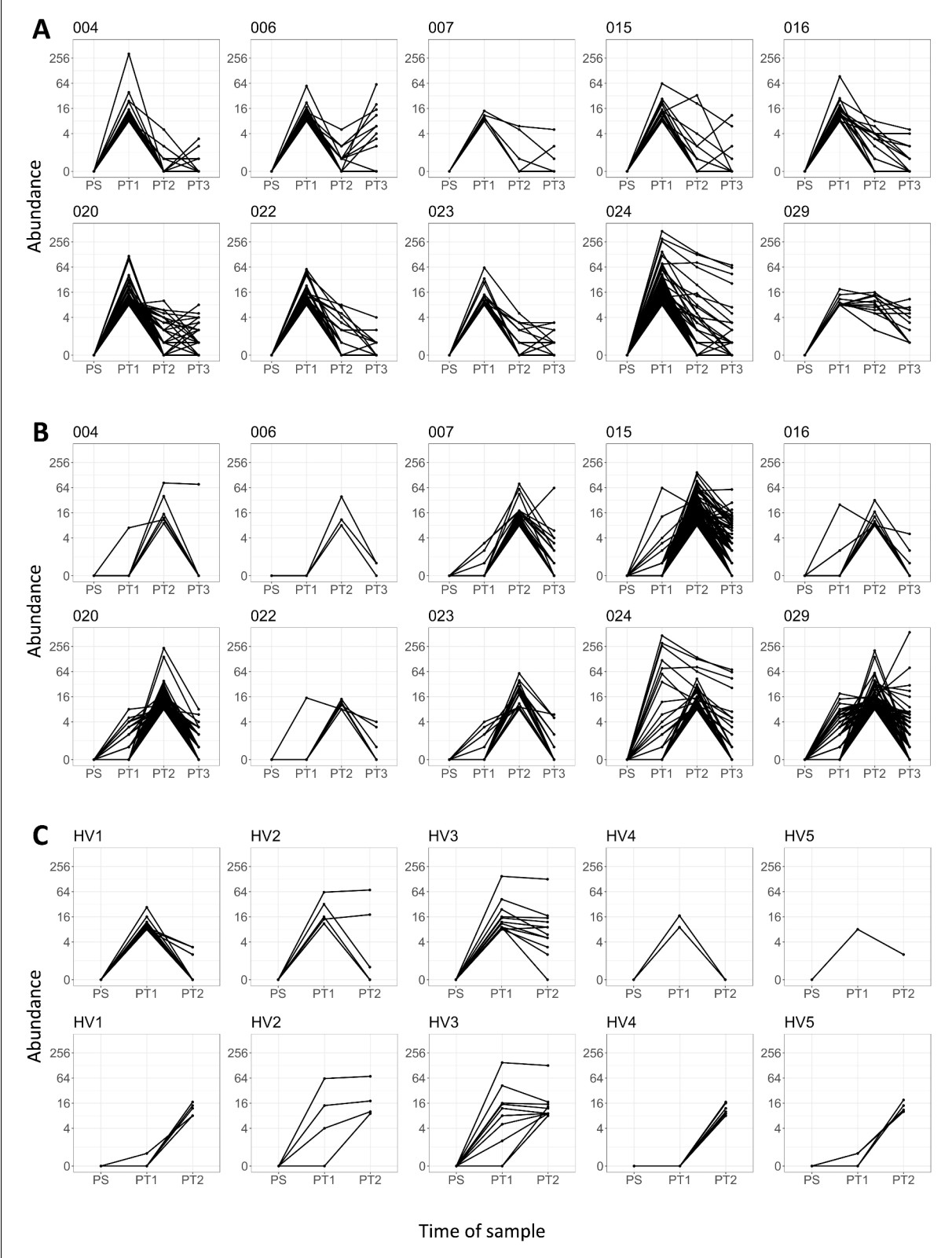

**Figure 3.** Dynamic changes in T cell receptor (TCR) frequency following sensitization. (**A**) The abundances of the PT1 expanded (threshold ≥8) beta TCRs at the four time points: PS, PT1, PT2, and PT3. Each panel is a different patient (n = 10). (**B**) The abundances of the PT2 expanded (threshold ≥8) beta TCRs at the four time points: PS, PT1, PT2, and PT3. Each panel is a different patient (n = 10). (**C**) Equivalent time points (0 weeks, 2 weeks, and 6 weeks) for five healthy volunteers. Top row is PT1 expanded beta TCRs; bottom row is PT2 expanded beta TCRs.

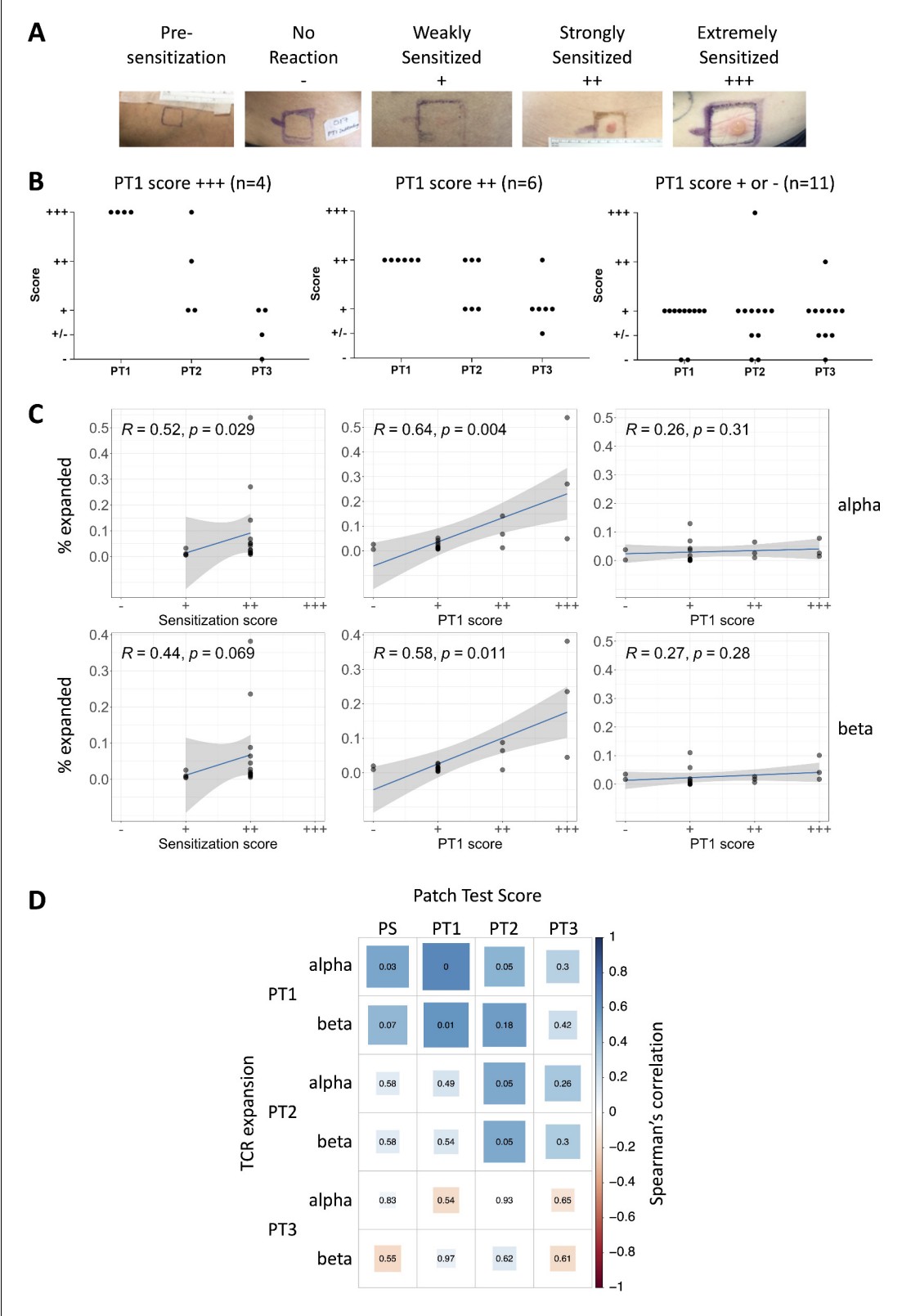

**Figure 4.** T cell receptor (TCR) expansion after exposure to diphenylcyclopropenone (DPC) correlates with the magnitude of allergic contact dermatitis. (**A**) Photographs showing examples of the varying levels of skin reaction observed in response to DPC; the reactions were classified according to standards set by the International Contact Dermatitis Research Group (no reaction [−], +, ++, +++). (**B**) The changes in patch test scores during treatment in patients with PT1 score of +++ (n = 4, left panel), ++ (n = 6, middle panel), and + or − (n = 11, right panel). (**C**) Left panels: The number of

*Figure 4 continued on next page*

Figure 4 continued

PT1 expanded (≥8) clones are plotted against the sensitization score. Center panels: The number of PT1 expanded (≥8) clones are plotted against the PT1 patch test score. Right panels: The number of PS expanded (≥8) clones are plotted against the patch test score at PT1. The blue line indicates the best fit linear model, with the model's 95% confidence intervals in gray. The inset text shows the Spearman's correlation coefficient, rho. (D) Correlation matrix representing the Spearman correlation between the number of PT expanded TCRs and the reaction recorded at the sensitization site or the patch test scores at each time point. The color and size of each square correspond to Spearman's rho, and non-adjusted p-values are shown.

suggesting some common driver of V/J gene usage, despite the fact that the individuals were not related.

We also looked for potential clustering of CDR3 sequences of the expanded TCRs (*Figure 6*). We have previously shown that amino acid triplets (sequences of three adjacent amino acids within the CDR3 region) can predict antigen specificity (*Sun et al., 2017*). We therefore compared the pairwise similarity of all expanded CDR3s using a metric that quantifies the number of shared triplets between two sequences, normalized for sequence length (an example of a string kernel in the machine learning literature *Shawe-Taylor and Cristianini, 2004*). We converted the matrix of pairwise similarities into a network (*Figure 6A*), by connecting all those TCRs with a similarity above a given threshold (75%). As control, we performed the same analysis on same-sized sets randomly sampled either from the combined pre-sensitization repertoire (*Figure 6B,C* (ii)) or from the combined repertoire of the healthy volunteers (*Figure 6B,C* (iii)). The expanded TCRs formed significantly more large clusters of 'related' TCRs than either control sets (*Figure 6B,C* (i)). We selected the largest cluster of CDR3 beta sequences (*Figure 6D*) and carried out an alignment (summarized in *Figure 6E*, full alignment in *Supplementary file 5*). The sequences showed a high level of similarity consistent with belonging to a set of TCRs with shared specificities (*Dash et al., 2017*; *Glanville et al., 2017*).

Taken together, the skewing of V and J genes, and the presence of clusters of similar TCRs were strongly suggestive that the expanded TCRs were enriched for TCRs responding to a limited number of specific epitopes.

## A dynamic Bayesian network can predict sensitization based on TCR sequence

The skewed V/J usage and the TCR clustering are indicative of an antigen-specific response. We therefore explored whether the repertoire of expanded TCRs was sufficiently distinct from that of the unselected repertoire so that it could be used to distinguish DPC-expanded TCRs or sets of TCRs from the unexpanded repertoire. We used a dynamic Bayesian network (DBN) (*Dagum et al., 1992*; *Murphy and Mian, 1999*; *Murphy, 2002*; *Pearl, 1988*; *Yao et al., 2008*), a model we have developed to interrogate antigen specificity in TCR sequences. The DBN is a probabilistic graphical model, which provides a flexible classification tool that can incorporate heterogeneous data (e.g. sequences as well as V/J usage).

We constructed a DBN to classify TCR beta CDR3 sequences into one of two classes: DPC-related CDR3s and randomly sampled pre-sensitization CDR3s from the same individuals. To construct the training/test sets for the model, we used the PT1 expanded CDR3 sequences (≥8) for the DPC class, which were at most 22 amino acids long (2019 beta sequences from 22 patients), and for the control class, a randomly generated set of the same number of TCRs from the combined pre-sensitization sequences. CDR3 sequences in the expanded set were excluded from the pre-sensitization (control) training set. The length of CDR3 sequences used in the model was capped at 22 in order to manage the computational complexity of the algorithm, while still utilizing all expanded sequences. To account for the variable length of the CDR3 sequences, the sequences were aligned by the first Cysteine of the CDR3 on the left, up to the final Phenylalanine of the CDR3, then 'completed' to length 22 using a dummy variable ('amino acid number 21'). The DBN took as input the amino acid in each position in the CDR3 (converted to an integer 0–21 for each amino acid or dummy), the V and J genes, as well as position-specific triplet scores and class scores (as explained in Materials and methods). The proposed probabilistic dependencies between these sequence features and antigen specificity are encoded into the network structure by directed edges (*Figure 7A*). Further details of the DBN and how it was trained and tested are provided in Materials and methods.

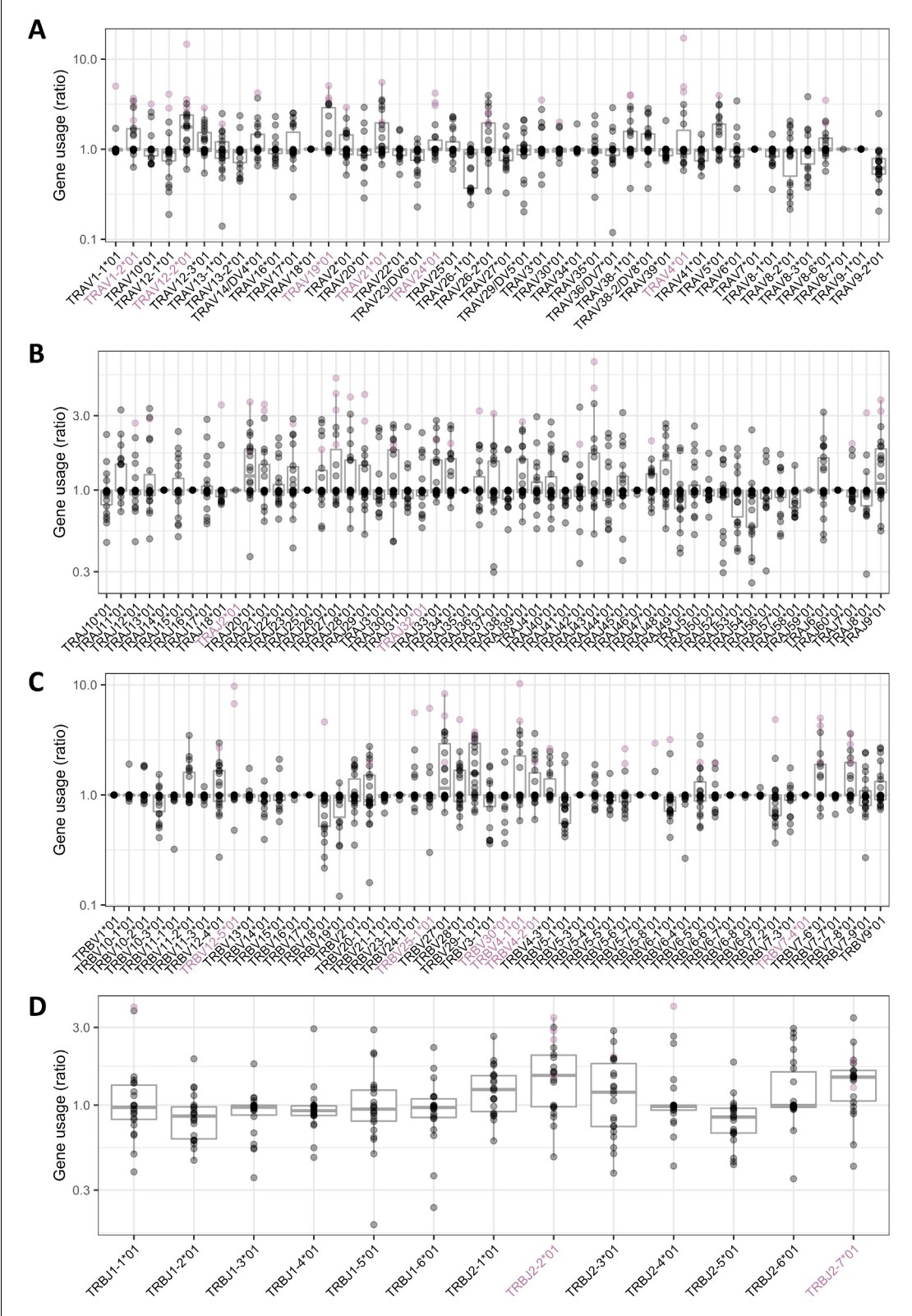

**Figure 5.** The expanded T cell receptors (TCRs) show characteristics of antigen-driven responses in their V and J gene usage. Relative frequency of V and J alpha and beta gene usage in the set of PT1 expanded TCRs (≥8) from 22 individuals, compared to the frequency in their respective pre-sensitization repertoires. Each dot is a patient. Genes significantly over-represented in the expanded set (see text for statistical test) are colored pink. Under-represented genes are not included in this figure, since the small number of expanded TCRs in some patients resulted in a large but less

*Figure 5 continued on next page*

*Figure 5 continued*

meaningful set of under-represented genes due to sampling. Genes colored pink on the x-axis are those that were significantly over-represented in the combined expanded set (1858 unique alpha TCRs and 2019 unique beta TCRs).

One of the key features of the DBN is that each 'slice' of the network is position specific. This enables modeling local as well as global sequence features. For example, *Figure 7B* demonstrates the relationship between neighboring amino acids in specific positions in the CDR3 sequence and antigen specificity. The probability of observing a specific amino acid in a particular position given its neighbors differs between DPC-expanded sequences and control sequences. Both over and under-represented pairs of amino acids can be seen, in particular in the middle of the CDR3. Such relationships are utilized by the model in classification.

The DBN classifier could classify the full DPC/control test sets with mean accuracy of 94% from 10 repeats of 10-fold cross validation, and sets of 10 sequences with mean accuracy 77.7% (*Figure 7C* upper panel). Model confidence in classifying individual sequences correctly, taken as the difference in the log likelihood between the DPC and the control models for each sequence, varied between different test sets and sequences. Where the DBN was asked to classify every sequence regardless of confidence level, the mean accuracy was 54.3%. When taking the 5% or 10% most confident assignments, individual sequences were classified with mean accuracy of 64.7% and 60.6% respectively, from 10-fold cross validation (*Figure 7C* lower panel). The DBN provides mechanistic insights into which features are critical for classification, in addition to confidence bounds on its decisions, since the probabilities used by the model in classification are transparent.

## Discussion

The results presented above report the first comprehensive analysis of the TCRrep in the context of controlled exposure to a contact sensitizer in humans. The key findings are that the response to sensitizer is accompanied by a dynamic, robust polyclonal increase in abundance of a defined set of TCR genes which presumably reflects clonal expansion of antigen-specific T cells in response to DPC. Notably, the breadth of the response, as captured by the number of TCRs that increase in frequency, is very diverse between individuals and predicts the degree of sensitization. Furthermore, the response of individual TCRs, which reflect individual T cell clonal dynamics, is transient, with most TCRs returning to low frequencies, or disappearing by the subsequent time point.

The immunological mechanisms that drive ACD have been studied in detail (see for example *Kimber et al., 2012*). The dose and frequency of exposure are both key parameters in determining the strength of the response (*Friedmann, 2006*; *Friedmann, 2007*), although it is clear that the full impact of different factors on the observed variance in the population is still not fully understood, and is one limitation in accurate prediction of allergy to chemical exposure in the context of consumer safety (*Basketter and Safford, 2016*). There is widespread agreement that activation of both CD4+ and CD8+ antigen-specific T cells plays key roles, although antibody may also be important (*Singleton et al., 2016*).

The nature of the molecular interaction between the sensitizer, which is usually a chemical hapten that reacts with macromolecules including proteins in the skin following exposure, remains much less well understood than for microbial or viral peptide antigens. There are no MHC tetramers or multimers which can be used to identify DPC or indeed any hapten-specific T cells. Nevertheless, a number of lines of evidence support our hypothesis that the TCRs that increase in frequency following sensitization represent an antigen-specific response. The observed changes from zero to eight are unlikely to be due to sampling effects; in fact, modeling sampling as a Poisson process indicates that changes from zero to eight or more have individual probabilities of <0.05. The observed changes are also unlikely to arise from exposure to unknown antigens (e.g. infectious agents) since the number of expanded TCRs at the first time point post-sensitization is greater than the number of expanded TCRs observed in unsensitized individuals sampled at similar time points, for the great majority of the patients tested. The functional link between TCR expansion and DPC exposure is also strengthened by the observed statistical correlation between the number of expanded TCRs and the vigor of sensitization, as measured by the patch test. Moreover, the expanded set of TCRs show strong evidence of V and J gene skewing, and of increased CDR3 clustering, both well-

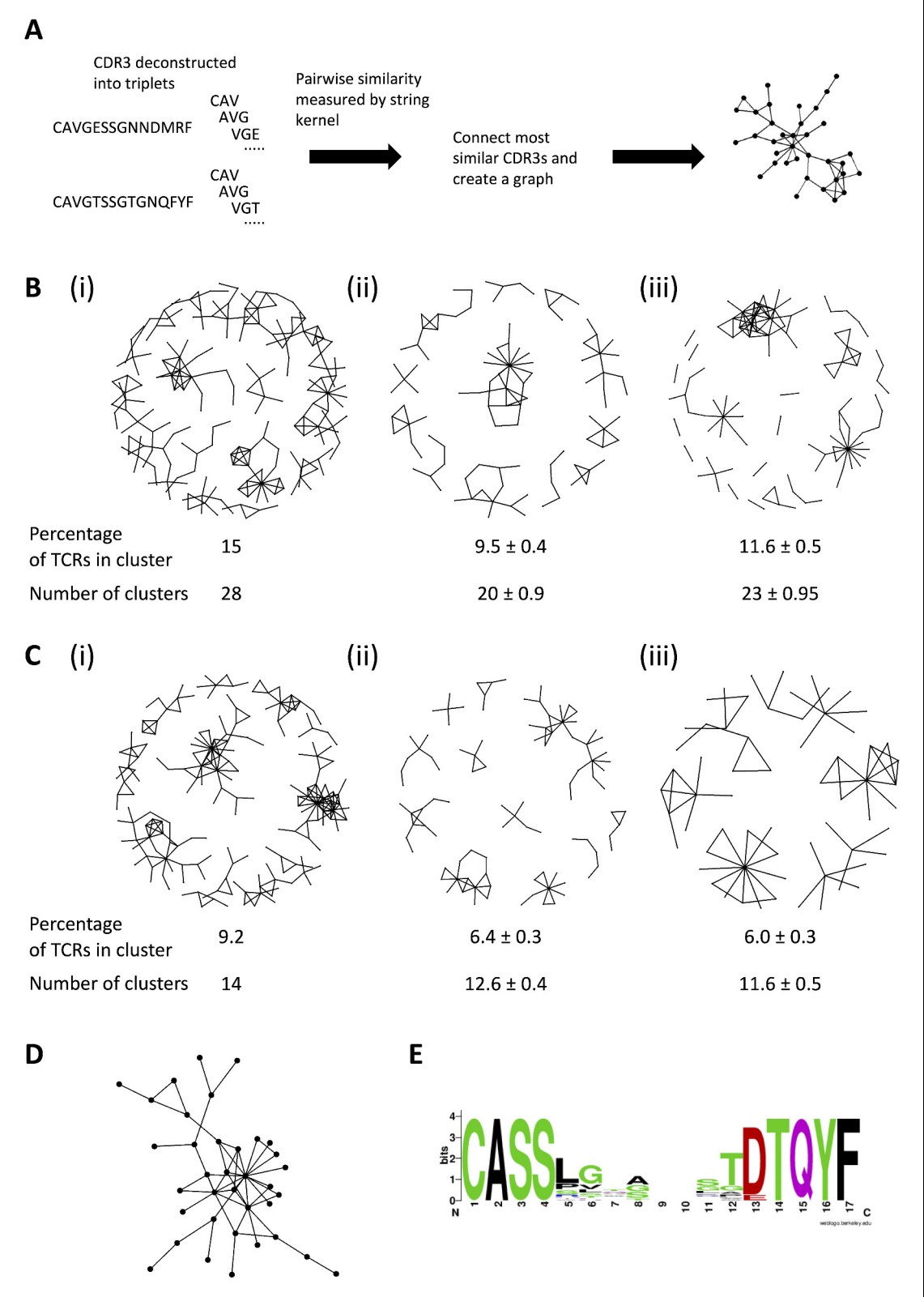

**Figure 6.** The expanded T cell receptors (TCRs) show clustering characteristics of antigen-driven responses. (A) Diagram illustrating the similarity graph construction process. Individual CDR3s are deconstructed into overlapping series of contiguous amino acid triplets, and the pairwise similarity between two CDR3s is calculated as the normalized string kernel. Two CDR3s which have a pairwise similarity of >0.75 are connected by an edge. (B) Clusters formed from the CDR3 sequences of (i) the PT1 expanded alpha TCRs ($\geq$ 8), (ii) an equal-sized (1858) set of alpha TCRs sampled randomly from the

*Figure 6 continued on next page*

Figure 6 continued

combined pre-sensitization repertoires of the same patients, and (iii) a size-matched set randomly sampled from the combined healthy volunteer alpha TCRs. The numbers under each network diagram show the percentage of the 1858 TCRs which are incorporated in a cluster, and the number of clusters. The numbers for the pre-sensitization and healthy volunteers show the average ± standard error of the mean for each parameter. (C) As (B), but for beta sequences (2019 expanded TCRs, or the same number of control TCRs). (D) The largest (35 unique CDR3s) cluster of PT1 expanded TCR beta sequences, from panel (C) (i). (E) An alignment of CDR3 sequences from the cluster shown in panel (D). The alignment is illustrated as a sequence logo (https://weblogo.berkeley.edu/logo.cgi). The full alignment is shown in *Supplementary file 5*.

established signatures of an antigen-specific T cell response (*Thomas et al., 2014*; *Sun et al., 2017*; *Davis et al., 1995*; *Glanville et al., 2017*; *Dash et al., 2017*). Finally, these features are shown to be discriminatory enough to allow classification of TCRs into DPC-related and pre-sensitization TCRs by the DBN. Taken together, this set of observations suggest that DPC, perhaps in the form of a set of modified-self peptide adjuncts, stimulates a transient expansion of T cells carrying a defined set of related TCRs. The observation that TCR clustering, V region enrichment, and DBN classification could be observed across a panel of unrelated individuals is particularly intriguing, and suggests that hapten-specific responses may be less dependent on MHC matching than conventional peptide-specific responses. However, further more detailed study is required to determine the molecular target of the DPC-responsive TCRs.

DPC is considered a potent sensitizer (*Stute et al., 1981*; *Mose et al., 2017b*). Nevertheless, we noted a wide range of quantitative responses in different individuals, at least as measured by standard patch test results. This was reflected in a wide range of expanded TCRs, and the number of expanded TCRs at the first post-sensitization sample (2 weeks) showed strong correlation with the strength of the response. The reasons for this inter-individual variation are unknown, but could reflect inter-individual differences in immune response (*Brodin et al., 2015*), or potentially differences in factors that control hapten penetration, such as skin thickness or composition (*Reynolds et al., 2019*). Understanding the factors that determine this variation may be important in developing novel approaches to ACD risk prediction (*MacKay et al., 2013*; *Maxwell and Mackay, 2008*).

In our study, low responders were not boosted by repeated exposure to DPC over many months. In fact, the response was usually maximal at the first patch test, suggesting that regulatory mechanisms come into play to limit the response in vivo. Similarly the initial TCR expansion did not continue to increase despite repeated exposure to antigen, and indeed the majority of TCRs showed very transitory responses, with individual TCR frequencies falling to around pre-sensitization frequencies by the next time point. Similar rapid expansion and subsequent contraction have been observed in response to live attenuated yellow fever vaccine (*DeWitt et al., 2015*; *Pogorelyy et al., 2019*; *Pogorelyy et al., 2018*), one of few examples where TCR repertoire has been studied before and after challenge in a human setting. The latter study (*Pogorelyy et al., 2018*) precisely documents changes in TCR frequency in three pairs of twins, at several time points before and after exposure to a single dose of the live attenuated YFV 17D vaccine. The study documented the expansion and then contraction of several hundred TCR genes, which reflected the clonal expansion, and subsequent contraction of a polyclonal set of vaccine-specific T cells. However, in this case antigen was delivered as a single dose of vaccine, which gives rise to systemic but very transitory viremia. T cell contraction in the vaccine context may therefore reflect the rapid disappearance of antigen. In contrast, the patients in the current study were exposed to repeated doses of DPC weekly over the course of many months for their therapeutic benefit. The mechanisms which limit the DPC response are therefore more likely to be due to intrinsic regulatory pathways (e.g. Treg induction), or changes in intrinsic migratory pathways rather than simply reflect antigen disappearance. We noted that a second wave of TCRs appeared at increased frequency at PT2, 6 weeks post-sensitization. This set of 'late' TCRs was almost completely distinct from the 'early' peak at week 2. Furthermore, there was no correlation between the magnitude of the patch test score at PT1 and the number of TCRs expanded at PT2. Additional experiments will be needed to explore whether this set of late TCRs represents part of a regulatory mechanism, which regulates and limits further T cell expansion during chronic exposure to DPC.

In conclusion, this study is the first analysis of in vivo TCR repertoire changes in response to a chemical allergen, and is one of only a handful of studies that document in vivo longitudinal changes

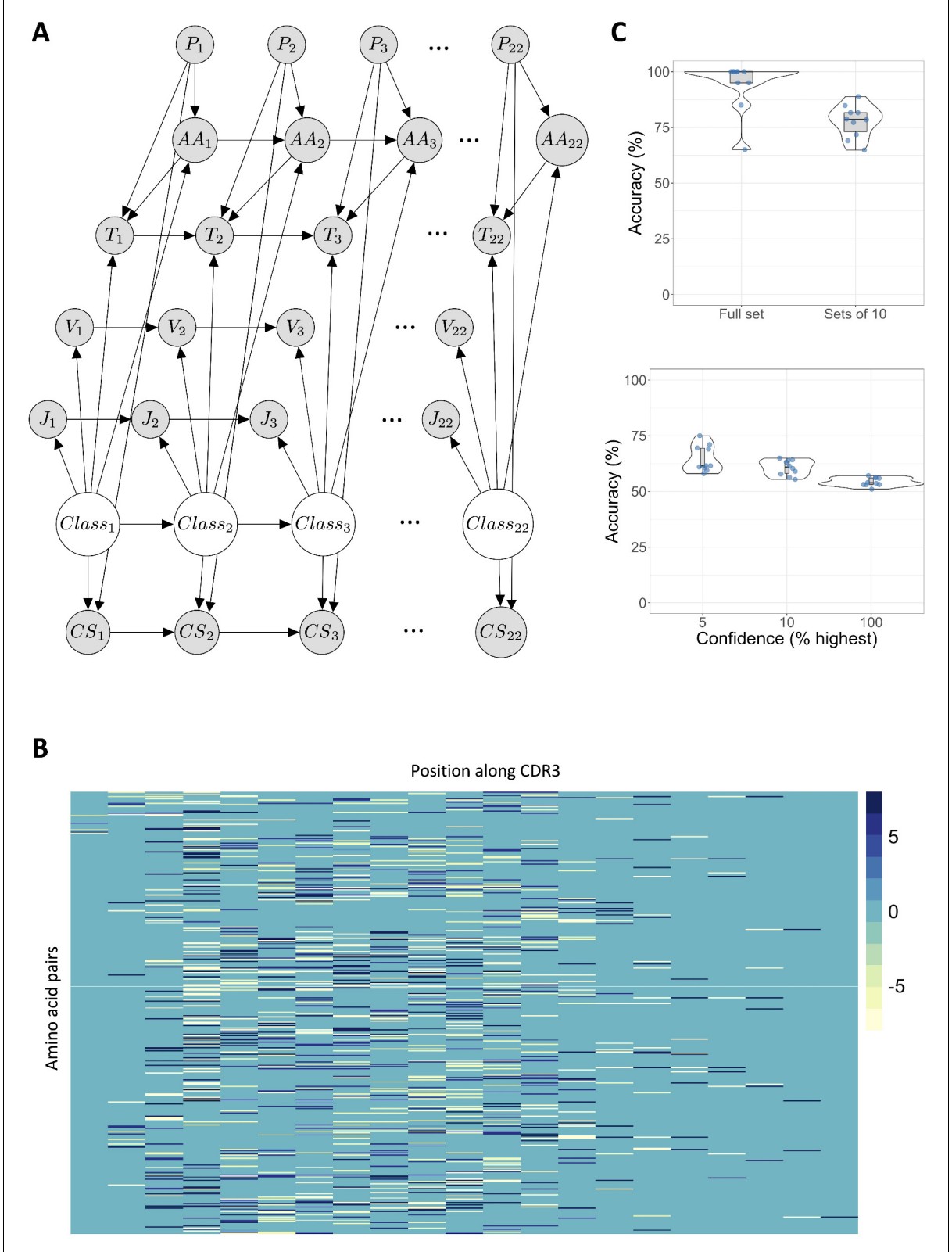

**Figure 7.** A dynamic Bayesian network (DBN) can predict sensitization based on T cell receptor (TCR) sequence. (**A**) The directed acyclic graph depicts the DBN structure, unrolled over 22 time slices (positions). Directed edges capture probabilistic dependencies. $P_i$ indicates position $i$ along the CDR3 sequence; $AA_i$ the amino acid in position $i$; $T_i$ is the triplet score for the triplet of amino acids in positions $\langle i-2, i-1, i \rangle$; $V_i$ and $J_i$ are the V and J genes (constant for each sequence); $CS_i$ the class score at position $i$, and $Class_i$ determines whether the sequence comes from the

*Figure 7 continued on next page*

*Figure 7 continued*

diphenylcyclopropenone (DPC)-related expanded set, or the control set. (**B**) The probability of observing pairs of neighboring amino acids differs between DPC-related sequences and control sequences from the pre-sensitization repertoire of the same individuals. Each column in the heatmap corresponds to a position along the CDR3 sequence, each row to an ordered pair of amino acids (total 400). If row $i$ in the heatmap corresponds to amino acid pair $\langle X, Y \rangle$, then position $\langle i, j \rangle$ in the matrix is the ratio of the probability of observing amino acid $Y$ in position $j+1$ in the expanded sequences against the probability of observing $Y$ in the same position in an equal-sized random sample of control sequences, given amino acid $X$ has been observed in position $j$. (**C**) The upper panel shows the mean classification accuracy for classifying the full DPC/control test sets, and sets of 10 DPC/control sequences. The lower panel shows mean classification accuracy for individual sequences. 5% confidence refers to the top 5% of sequences with greatest log likelihood difference between the models, and similarly for 10%. 100% is the accuracy when classifying every sequence. Each dot depicts the mean of one of the train-test sets from 10-fold cross validation over 10 generations of the model. Each training set consisted of 3618 expanded sequences and control sequences in equal proportions, and the test sets 402 sequences in equal proportions (201 expanded and 201 control).

The online version of this article includes the following source data for figure 7:

**Source data 1.** Table of expanded T cell receptors (TCRs).
**Source data 2.** Table of control T cell receptors (TCRs).

in TCRrep in response to immunization in humans. We cannot rule out that the immune response in this population is different from healthy individuals, since all patients suffered from some form of alopecia, which is believed to have an autoimmune etiology (*Trüeb and Dias, 2018*). Nevertheless, DPC is unlikely to bear any relation to the autoimmune target in alopecia, since similar therapeutic responses (hair growth) can be observed in patients with alopecia treated with other completely unrelated contact sensitizers. While the qualitative balance of immune response may be influenced by the underlying autoimmune background, the specificity of the TCR repertoire is likely to reflect fundamental features of the response to chemical haptens. The study confirms our previous in vitro findings that the response even to a simple chemical such as DPC is polyclonal involving dozens or even hundreds of TCRs. The study also highlights the dynamic nature of the TCR repertoire. Further studies will be required to unravel the complex mechanisms that regulate the immune response to chronic antigen exposure.

# Materials and methods

## Key resources table

| Reagent type (species) or resource | Designation | Source or reference | Identifiers | Additional information |
| --- | --- | --- | --- | --- |
| Software, algorithm | Decombinator V4 | https://github.com/innate2adaptive/Decombinator | RRID:SCR_006732 | This software suite is under active development; latest versions available at the GitHub site. |
| Other | TCRseq protocol | *Uddin et al., 2019b* PMID:31727254 | | This protocol is in a continuous state of development. The full details of the current stable version including primer sequences, PCR conditions etc. are all in the attached reference. For latest development contact the corresponding author on b.chain@ucl.ac.uk |

## Patient recruitment

A total of 34 patients were recruited to this study (NRES Ethics Committee East of England – Cambridgeshire and Hertfordshire [14/EE/1067]). Participants were recruited from patients who had been diagnosed with alopecia, were aged between 18 and 70, identified as suitable for DPC treatment by a consultant dermatologist, and were now attending their first visit to the Alopecia Clinic at Salford Royal Hospital for DPC therapy. This study ran alongside patients' prescribed DPC treatment (weekly doses of DPC to the scalp to induce inflammation and hair regrowth). The study timeline in terms of treatment and sample collection is provided in *Figure 1A*. All participants gave their informed consent to participate and were free to withdraw from the study at any time and for any

reason without affecting their treatment. Patients were excluded from the study if they were pregnant.

Twenty-nine of the individuals who participated in the study provided blood samples for TCR sequencing (TCRseq), for between one and four of the study time points (pre-sensitization, and at 2, 6, and 24 weeks of DPC treatment). Flow cytometry data was obtained for peripheral blood mononuclear cells (PBMCs) from 10 treated patients, and patch test data for 24 patients. The clinical response to treatment (in terms of hair regrowth) and the associated immunological changes will be discussed in a separate publication. A summary of the patient demographics and the samples collected is shown in *Table 1*.

As controls for the TCR sequencing, five healthy volunteers were bled three times, at day 0, 2 weeks, and 6 weeks. All subjects gave written informed consent in accordance with the Declaration of Helsinki. The protocol was approved by the University College London Hospital Ethics Committee 06/Q0502/92.

## Sensitization and patch testing

Sensitization to DPC was induced by application of 2% DPC (in acetone) to a 2 cm by 2 cm patch of skin on the upper inner arm. Sensitization at the site of application was assessed 14 days later and scored as described below (sensitization score). Patch testing at a remote site (upper back) was conducted 2 weeks (PT1), 6 weeks (PT2), and 24 weeks (PT3) after application of the sensitizing dose of DPC. Patch testing was performed on the skin of the upper back using Finn chambers (8 mm inner diameter) containing 0.01% DPC in acetone. After 6 days, patients' reactions were scored as no reaction (−), weakly sensitized (+), strongly sensitized (++), or extremely sensitized (+++) according to standards set by the International Contact Dermatitis Research Group.

## TCR sequencing

The α and β chains of the TCR repertoire of 29 participants were sequenced using a method that starts with total RNA isolated from unfractionated whole blood, collected in Tempus Blood RNA tubes (Thermofisher #4342792) using the manufacturer's protocol for RNA extraction. The pipeline introduces unique molecular identifiers attached to individual cDNA molecules to provide a quantitative and reproducible method of library preparation. Full details for both the experimental TCRseq library preparation and the subsequent computational analysis (V, J, and CDR3 annotation) using Decombinator are published in *Oakes et al., 2017b*; *Uddin et al., 2019a*.

## Flow cytometry

PBMCs were isolated from 30 mL whole blood (diluted 1:1 in PBS) from 10 patients layered over an equal volume of Histopaque 1077 (Sigma Aldrich) according to the manufacturer's instructions. Cells were stored at a concentration of $5 \times 10^7$ cells/mL in 10% DMSO/90% human AB serum (Sigma Aldrich) at −80°C until required.

Flow cytometric analyses were performed on previously frozen cells using antibodies obtained from eBioscience, with single-stained controls used for compensation. Naïve and memory CD4+ and CD8+ T-cell subsets were identified as previously described (*Appay et al., 2008*). Naïve cells were defined as CD45RA+ and CD27+; central memory as CD45RA− CD27+; effector memory as CD45RA− CD27−, and EMRA as CD45RA+ CD27−. Flow cytometry was performed on a minimum of 10,000 cells using a FACS-calibur (Becton Dickinson, Mountain View, CA) and data analysis performed in FlowJo (TreeStar) using standardized gating across all samples.

## Statistical and mathematical analysis

Statistical analyses were performed using the statistical programming language R [R version 4.0.2 (2020-06-22)]. Mann–Whitney U-tests were used to compare between two unmatched groups and paired t-tests or Wilcoxon signed-rank tests between two paired groups. All statistical comparisons between more than two groups were done using Kruskal–Wallis non-parametric tests with post hoc Dunn Test and Benjamini–Hochberg correction for multiple testing. Statistical significance in all tests was accepted above the 0.05 threshold.

To calculate which V and J genes were significantly over- or under-expressed in the DPC-expanded TCR sets, at both the individual patient level and for the entire expanded set, we

generated 2*1000 independent random samples of equal-sized sets from the pre-sensitized individual/combined population, calculated the ratio between the two, and compared this with the ratio derived from the set of interest against a further 1000 random simulations. A significance level of 0.05 was then achieved for genes that ranked in either the top 50 'ratios' (under-represented) or bottom 50 in the list (over-represented).

## TCR clustering

The CDR3 protein sequences of expanded TCRs were identified using the package CDR3translator (https://github.com/innate2adaptive/Decombinator). The pairwise similarity between TCRs was measured on the basis of amino acid triplet sharing, which was calculated using the normalized string kernel function stringdot (with parameters stringdottype='spectrum', length = 3, normalized=TRUE) from the Kernlab package (*Karatzoglou et al., 2004*). The kernel was calculated as the number of amino acid triplets (sets of three consecutive amino acids) shared between two CDR3s, normalized by the number of triplets in each CDR3 being compared. The TCR similarity matrix was converted into a network diagram using the iGraph package in R (*Csardi and Nepusz, 2006*). Two TCRs were considered connected if the similarity index was above 0.75. A range of thresholds were explored, and the lowest threshold that consistently gave few large (>3 nodes) clusters using random samples of TCRs from the study was chosen. The sequences from individual clusters were aligned using Aliview (*Larsson, 2014*), and the consensus visualized using webLogo (https://weblogo.berkeley.edu/logo.cgi).

## TCR repertoire classification using a DBN

DBNs are a type of probabilistic graphical model consisting of a directed acyclic graph and a set of conditional probability distributions. The probability of the set of variables (nodes) of the system $X_t$ at time $t$ given its state at time $t-1$ can be calculated by

$$P(X_t|X_{t-1}) = \prod_{i=1}^{N} P\left(X_t^{(i)} \mid \mathrm{Pa}\left(X_t^{(i)}\right)\right)$$

where $N$ is the number of variables, $X_t^{(i)}$ is the $i$'th node in time slice $t$, and $Pa(X_t^{(i)})$ are the parents of $X_t^{(i)}$. The TCR DBN was built in MATLAB (2019b), using the Bayes Net Toolbox (*Murphy, 2001*). The DBN took as input the position along the CDR3 sequence, the amino acid in each position, the V and J genes, a position-specific triplet score, and a class score relative to sequence position. An edge between neighboring amino acids was included in the network to model the observed non-independence between neighboring amino acids in their specific sequence positions and the sequence specificity. The triplet score was calculated by taking for each position the triplet containing the two previous amino acids in the CDR3 and ranking it against other triplets in this position in the control and DPC training sets. The triplet was given a score of 1 if it appeared more frequently (in a given position) in the DPC training set than in the control training set, which consisted of the same number of sequences randomly sampled from the pre-sensitization repertoires of the same patients. Triplets that were more frequent in the control set were scored 2, and triplets that appeared equally in the DPC and control sets were scored 1 or 2 uniformly at random. Positions 1 and 2 were taken as a singleton and an ordered pair of amino acids respectively and scored similarly. Triplets were chosen since they were sufficiently high dimensional to capture amino acid dependencies but remained computationally feasible. To calculate the class scores, the following procedure was followed: 1. Each V gene in the training set was scored 1 if it was more prevalent in the DPC set, 2 if in the control set, and 1 or 2 uniformly at random otherwise. 2. The J genes were scored similarly. 3. For each sequence position, every amino acid (and an additional dummy amino acid added to the end of CDR3s shorter than 22 to equalize sequence lengths) was scored 1 or 2 as above. 4. Finally, the class score for position $i$ was assigned 1 if the total number of '1' scores obtained by Steps 1–3 and the triplets scores from positions $1,2,...,i-1$ was greater than the number of '2' scores; a score of 2 if the number of '2's was greater, and 1 or 2 randomly otherwise. Training sequences were assigned 1 or 2 in position 22 according to their original class. This process ensured knowledge about the likelihood of belonging to either set could be learned by the model across time slices (positions). Class scores for the test sets were calculated based on gene, amino acid, and triplet frequencies in the training sets. Nodes depicting specificity (DPC or control) were included as

hidden (latent) variables (the Class nodes in *Figure 7A*). Inference on the network was done using the Junction Tree algorithm, and parameters were initiated with Dirichlet priors and fitted to the network by Expectation Maximization with a maximum of 10 iterations. Two DBNs were constructed, one for the DPC training set and one for the control set. To classify TCR sequences, the log likelihood of each model was calculated. For sets of sequences, we calculated the sum log likelihood for each model. The results of the DBN were evaluated using 10-fold cross-validation, running each train-test pair 10 times, and calculating the mean classification accuracy (number of sequences/sets of sequences correctly classified divided by total number of sequences/sets of sequences). The training and test sets all consisted of equal numbers of DPC-related and control sequences, and running the DBN with shuffled class labels returned ~50% accuracy. The network diagram in *Figure 7A* was created using the LaTeX TikZ BayesNet package (https://github.com/jluttine/tikz-bayesnet; *Luttinen and Dietz, 2013*).

---

## Additional information

### Competing interests
Gavin Maxwell: GM is an employee of Unilever PLC. Apart from GM's contribution (see author contributions), the funder was not involved in the study design, collection, analysis, and interpretation of data, the writing of this article or the decision to submit it for publication. The other authors declare that no competing interests exist.

### Funding

| Funder | Author |
|---|---|
| Unilever | Benny Chain |

The funders had no role in study design, data collection and interpretation, or the decision to submit the work for publication.

### Author contributions
Tahel Ronel, Conceptualization, Software, Formal analysis, Visualization, Methodology, Writing - original draft; Matthew Harries, Supervision, Investigation, Project administration, Writing - review and editing; Kate Wicks, Helen Singleton, Investigation, Writing - review and editing; Theres Oakes, Investigation, Methodology, Writing - review and editing; Rebecca Dearman, Resources, Supervision, Writing - review and editing; Gavin Maxwell, Conceptualization, Funding acquisition, Project administration, Writing - review and editing; Benny Chain, Conceptualization, Formal analysis, Supervision, Funding acquisition, Writing - original draft, Project administration

### Author ORCIDs
Tahel Ronel [ID] https://orcid.org/0000-0002-9513-9181
Benny Chain [ID] https://orcid.org/0000-0002-7417-3970

### Ethics
Human subjects: The protocol was approved by the University College London Hospital Ethics Committee 06/Q0502/92. A total of 34 patients were recruited to this study (NRES Ethics Committee East of England - Cambridgeshire and Hertfordshire [14/EE/1067]). Participants were recruited from patients who had been diagnosed with alopecia, were aged between 18 and 70, identified as suitable for DPC treatment by a consultant dermatologist, and were now attending their first visit to the Alopecia Clinic at Salford Royal Hospital for DPC therapy. This study ran alongside patients' prescribed DPC treatment (weekly doses of DPC to the scalp to induce inflammation and hair regrowth). All participants gave their informed consent to participate, and were free to withdraw from the study at any time and for any reason without affecting their treatment. Patients were excluded from the study if they were pregnant.

Decision letter and Author response
Decision letter https://doi.org/10.7554/eLife.54747.sa1
Author response https://doi.org/10.7554/eLife.54747.sa2

## Additional files

**Supplementary files**

• Supplementary file 1. Unique and total TCR numbers for each TCRseq sample. '_TRA' are the TCR alpha chain samples, and '_TRB' the beta chain samples.

• Supplementary file 2. Repeated exposure to DPC does not alter the global structure of the peripheral blood TCR repertoire. (A) The Shannon diversity index of the healthy volunteers (n = 14 samples from five individuals), pre-sensitization (n = 25), and post-sensitization (n = 58; from all three time points) TCR repertoire samples. All samples were randomly subsampled to the minimum sample size (20,172 alpha TCRs), and the Shannon diversity index of the subsample was then calculated. Each sample is represented by a dot. The box plots show the median, and lower and upper quartiles of each group. Differences in the distribution of the three groups were tested using a Kruskal–Wallis rank sum test and were non-significant (p=0.7658). (B) The Gini inequality coefficient of the healthy volunteers, pre-sensitization and post-sensitization alpha chain TCR repertoire samples, subsampled as in (A). Differences in the distribution of the three groups were tested using a Kruskal–Wallis rank sum test and were non-significant (p=0.9062). (C) The number of alpha TCRs that appear with a frequency of 1/1000 or higher in each sample (termed 'abundant TCRs'), for the healthy volunteers, pre-sensitization and post-sensitization samples, subsampled as in (A) and (B). A Kruskal–Wallis rank sum test revealed no statistical difference between the groups (p=0.9256). (D) – (F) The alpha chains of the sensitized samples were separated according to time point: PT1 (n = 23), PT2 (n = 18), and PT3 (n = 17). The Shannon diversity index (D), the Gini coefficient (E), and the number of abundant clones (F) of these subsamples were calculated. Kruskal–Wallis rank sum tests were used to compare between the three groups in each case. All tests showed no statistically significant difference, with p=0.9608, p=0.9281, and p=0.8681 respectively.

• Supplementary file 3. Sensitization with DPC induces a transient expansion in the frequency of a small subset of the TCR repertoire. (A) The abundance distribution of TCRs at PS and PT1. All samples were subsampled to the same number of TCRs (28,000). Each unique TCR is represented by a dot, and the axes represent the number of times it is observed in the PS (x-axis) and PT1 (y-axis) sample of the same individual, and equally spaced time points for the healthy volunteers. The pink dots identify a population of TCRs absent in the PS sample and expanded (abundance ≥8) in the PT1 sample. The blue dots identify a population of TCRs absent in the PT1 sample and expanded (≥8) in the PS sample. The numbers indicate the percentage of PT1 expanded TCRs (pink) and PS expanded TCRs (blue). (B) The correlation between the percentage of PT1 expanded alpha chain TCRs (x-axis) and the percentage of PT1 expanded beta chain TCRs (y-axis) for each individual (n = 22), subsampled as in (A). Spearman's rho = 0.85, p<0.0001. The line x=y is shown.

• Supplementary file 4. Dynamic changes in TCR frequency following sensitization. (A) The abundances of the PT1 expanded (threshold ≥8) alpha TCRs at the four time points: PS, PT1, PT2, and PT3. Each panel is a different patient (n = 10). (B) The abundances of the PT2 expanded (threshold ≥8) alpha TCRs at the four time points: PS, PT1, PT2, and PT3. Each panel is a different patient (n = 10). (C) Equivalent time points (0 weeks, 2 weeks, and 6 weeks) for four healthy volunteers for whom we had all three times points. Top row is PT1 expanded alpha TCRs; bottom row is PT2 expanded alpha TCRs. The fourth volunteer had no PT1 expanded alpha TCRs. The fifth volunteer had no PT1 expanded alpha TCRs and no PT2 alpha sample.

• Supplementary file 5. Sequence alignment of the CDR3 sequences from the largest cluster of TCR beta PT1 expanded CDR3s. The CDR3s of the largest cluster of PT1 expanded CDR3 beta sequences (see *Figure 6D and E*) were aligned using the MUSCLE alignment algorithm in Aliview (https://ormbunkar.se/aliview/).

• Transparent reporting form

## Data availability

All DNA sequences have been submitted to the Sequence Read Archive under identifier PRJNA592875.

The following dataset was generated:

| Author(s) | Year | Dataset title | Dataset URL | Database and Identifier |
|---|---|---|---|---|
| Ronel T, Harries M, Wicks K, Oakes T, Singleton H, Dearman R, Maxwell G, Chain B | 2019 | T cell receptor sequencing of alopecia patients during skin sensitisation | https://www.ncbi.nlm.nih.gov/sra/PRJNA592875 | NCBI Sequence Read Archive, PRJNA592875 |

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
