## [Decision Letter]

**Acceptance summary:**

This manuscript presents a longitudinal analysis of the whole blood Tcell repertoire dynamics in patients with Alopecia Areata upon primary, secondary and chronic exposures to an allergic chemical agent. The authors have identified signatures of polyclonal T-cell response following sensitization involving stereotypical T-cell receptor genes. Aside from the specific case of T-cell response to the allergic agent in this study, the developed methods could be used for the general and important problem of identifying reactive Tcell populations to antigenic challenges.

**Decision letter after peer review:**

Thank you for submitting your article "The clonal structure and dynamics of the human T cell response to an organic chemical hapten" for consideration by *eLife*. Your article has been reviewed by three peer reviewers, and the evaluation has been overseen by a Reviewing Editor and Aleksandra Walczak as the Senior Editor. The following individual involved in review of your submission has agreed to reveal their identity: Yuval Elhanati (Reviewer #2).

The reviewers have discussed the reviews with one another and the Reviewing Editor has drafted this decision to help you prepare a revised submission.

This manuscript presents a longitudinal analysis of the whole blood TCR repertoire dynamics in patients with Alopecia Areata upon primary, secondary and chronic exposure to DPC treatment. The authors have identified signatures of polyclonal T-cell response following sensitization involving stereotypical T-cell receptor genes. Using a Dynamic Bayesian Network, the authors characterize a sequence based classifier that with a high accuracy can predict sensitization based on TCR sequence. Data collection and analysis are performed carefully and the results are presented clearly, but there are still some concerns that we would like to see addressed.

1) Most of the analysis rely on recognizing the expanded clones. For this purpose TCRs were counted based on their fold change – 2, 4, 8 and 16. However, the fold change can be very sensitive to the sample size at each time point – smaller samples tend to have lower diversity, which translates into bigger clone frequencies. This bias in particular, is present for TCR clones with low multiplicities (i.e., the majority of TCRs in your data). For example, consider two replicate blood samples such that one is sequenced ten times deeper than the other. A lot of singletons (TCRs with count 1) should be present in the deep sample, whereas almost all of them should be absent in the other sample. To estimate frequencies however, authors adjust these absent counts from 0 to 1 and divide the adjusted counts by the total counts in each sample. As a result, these fluctuations due to sample depth would appear as 10-fold difference in frequency of the singleton clones between samples.

– To gauge the validity of the analysis, it would be insightful if authors show more basic statistics for the fold change in clone sizes: what are pre-sensitization frequency and counts for expanding clones? Are there a lot of clonotypes that grow from 0 to 8 counts? What fraction of pre-sensitization high-frequency clones expand? One way to show this is to make scatterplots for clone counts in PS vs counts in PT1, ideally in log-scale; see DeWitt et al., 2015 Figure 2 as an example.

– Given the concerns above, it is also necessary that authors provide a table with details on sample size in data (e.g. unique clonotypes and UMIs for each sequenced sample) and show that the results are robust to these differences. For example, do observed differences in Figure 2 hold, if all time points are down-sampled to the same number of UMI per sample?

2) A more detailed description of the methods for analysis of clone size dynamics is necessary: Are clonal frequencies estimated after adding 1 to zero counts or are missing clones assigned a minimum frequency in a given sample (see the example in comment 1)?

3) From the comparison of Figure 2 and Supplementary file 2 it seems that expansion analyses in α and β repertoires do not quite agree. In particular, for the α chain, there are no significant differences between control and patient groups. If there are strong clonal expansions, one expects a correlated fold increase in α and β chains that belong to the same clone. Can authors explain this discrepancy? Is there a correlation between the number of expanded clones in α and the β repertoires?

4) Presentation of Figure 3 should be improved. For examples, clonal dynamics can be shown by connected lines over time. For comparison, it would be useful to show similar statistics in healthy individuals.

5) Figure 4 is difficult to read, specifically the left panels in C, D and E. These plots highlight the dependence of the number of expanded TCRs on the expansion fold change, which is rather trivial and not the point the authors are making here. On the other hand, the differences between the different sensitization groups are difficult to see. A different visualization should be considered.

6) Figure 5: Although V/J usage can reflect a response to DPC, the authors should first analyze whether the V/J usage is patient dependent or treatment dependent. For instance, this can be done by performing the same analysis on all clonotypes from patients versus healthy volunteers, expanded clonotypes in healthy individuals versus 1000 random clones in healthy individuals, etc… In addition, the use of box-plot is not readable for all the V and J. Better summarizing the information as heatmap.

7) For DBN, it is not clear from the Materials and methods how the sequences are aligned since TCRs have variable length.

8) To train a DBN, the authors construct the negative set from TCRs of healthy donors which were not in the expanded positive set. A natural negative set would be the patient-specific TCRs that were present in the pre-sensitized sample but did not expand. Authors should check how their results are robust to this choice of negative set.

Also, TCRs are somewhat individual-specific and so using a negative set from different donors might be problematic and create an "easy" task for classification. This can be fixed by using donor specific training sets.

What is the ratio of positive to negative set? Again, how robust is the performance of DBN to this training ratio. Authors should provide access to their training and test sets as a separate supplementary file.

9) Can authors discuss in more detail the TCR features that are informative for DPC classification. Do the authors expect same classifier performance for other tasks (e.g. distinction of TCRs specific for certain viral epitope from TCRs specific for other epitopes), or high accuracy of DBN is limited to the special case of DPC-recognizing TCRs?

---

## [Author Response]

This manuscript presents a longitudinal analysis of the whole blood TCR repertoire dynamics in patients with Alopecia Areata upon primary, secondary and chronic exposure to DPC treatment. The authors have identified signatures of polyclonal T-cell response following sensitization involving stereotypical T-cell receptor genes. Using a Dynamic Bayesian Network, the authors characterize a sequence based classifier that with a high accuracy can predict sensitization based on TCR sequence. Data collection and analysis are performed carefully and the results are presented clearly, but there are still some concerns that we would like to see addressed.1) Most of the analysis rely on recognizing the expanded clones. For this purpose TCRs were counted based on their fold change – 2, 4, 8 and 16. However, the fold change can be very sensitive to the sample size at each time point – smaller samples tend to have lower diversity, which translates into bigger clone frequencies. This bias in particular, is present for TCR clones with low multiplicities (i.e., the majority of TCRs in your data). For example, consider two replicate blood samples such that one is sequenced ten times deeper than the other. A lot of singletons (TCRs with count 1) should be present in the deep sample, whereas almost all of them should be absent in the other sample. To estimate frequencies however, authors adjust these absent counts from 0 to 1 and divide the adjusted counts by the total counts in each sample. As a result, these fluctuations due to sample depth would appear as 10-fold difference in frequency of the singleton clones between samples.

We agree with this very important point. Indeed, the estimation of ratios is fraught with potential confounders of the type the reviewers raise. On the basis of the suggestions below, we have re-examined the raw data (what the reviewers refer to as “basic statistics”). We note (see below, and new Figure 2) that actually almost all the repertoire activity we observe following sensitisation is focused on TCRs ABSENT pre-sensitisation. Rather than calculate frequencies, and impute missing values, we therefore now focus on those TCRs absent at PS, and expanded at PT1. For comparison, we also analyse the complementary group – those absent at PT1, and expanded at PS. In this way, we do not have to impute any missing values, do not have to calculate fold expansion, but can focus directly on the proportion of TCRs present at an expanded threshold at a particular timepoint. In all quantitative analyses we subsample all the repertoires to the same number of TCRs (defined by unique UMIs), so as to add additional robustness to the analyses.

– To gauge the validity of the analysis, it would be insightful if authors show more basic statistics for the fold change in clone sizes: what are pre-sensitization frequency and counts for expanding clones? Are there a lot of clonotypes that grow from 0 to 8 counts? What fraction of pre-sensitization high-frequency clones expand? One way to show this is to make scatterplots for clone counts in PS vs counts in PT1, ideally in log-scale; see DeWitt et al., 2015, Figure 2 as an example.

We thank the reviewer for this excellent suggestion. This way of visualisation is indeed very instructive and these plots are included in Figure 2, and in Supplementary file 3.

– Given the concerns above, it is also necessary that authors provide a table with details on sample size in data (e.g. unique clonotypes and UMIs for each sequenced sample) and show that the results are robust to these differences. For example, do observed differences in Figure 2 hold, if all time points are down-sampled to the same number of UMI per sample?

We include unique TCRs and total UMIs for each sample in Supplementary file 1. We show that the proportion of expanded TCRs is not correlated to the number of UMIs per sample. We also carry out repeat subsampling of the data to the lowest number of UMIs and calculate the correlation between expansion and patch score. The qualitative picture is the same, although as discussed in more detail in the manuscript, the expansion threshold at which the mean correlation becomes statistically significant changes a little bit.

2) A more detailed description of the methods for analysis of clone size dynamics is necessary: Are clonal frequencies estimated after adding 1 to zero counts or are missing clones assigned a minimum frequency in a given sample (see the example in comment 1)?

This issue is no longer relevant since, as discussed above, we do not use fold change any more, and do not have to assign values to missing points.

3) From the comparison of Figure 2 and Supplementary file 2 it seems that expansion analyses in α and β repertoires do not quite agree. In particular, for the α chain, there are no significant differences between control and patient groups. If there are strong clonal expansions, one expects a correlated fold increase in α and β chains that belong to the same clone. Can authors explain this discrepancy? Is there a correlation between the number of expanded clones in α and the β repertoires?

Expanded α and β TCR numbers are indeed closely correlated (rho – 0.85, p-value<0.0001) (now included in Supplementary file 3). Occasionally there are small differences in statistics for α and β; we presume this reflects sampling effects or some other technical variation between samples.

4) Presentation of Figure 3 should be improved. For examples, clonal dynamics can be shown by connected lines over time. For comparison, it would be useful to show similar statistics in healthy individuals.

We have redrawn these clonal dynamics as lines and included healthy volunteers.

5) Figure 4 is difficult to read, specifically the left panels in C, D and E. These plots highlight the dependence of the number of expanded TCRs on the expansion fold change, which is rather trivial and not the point the authors are making here. On the other hand, the differences between the different sensitization groups are difficult to see. A different visualization should be considered.

We have completely redrawn this figure, focusing on the correlation between number of expanded TCRs and sensitization scores. We believe the results are now clear.

6) Figure 5: Although V/J usage can reflect a response to DPC, the authors should first analyze whether the V/J usage is patient dependent or treatment dependent. For instance, this can be done by performing the same analysis on all clonotypes from patients versus healthy volunteers, expanded clonotypes in healthy individuals versus 1000 random clones in healthy individuals, etc… In addition, the use of box-plot is not readable for all the V and J. Better summarizing the information as heatmap.

We agree with the reviewers that V/J usage may be patient dependent. For this reason, we have compared the expanded set of each individual against unexpanded sets from the same individual. This shows that certain genes are over-represented in the expanded TCRs compared to background for the individual. We have rearranged the plots to make them more readable.

7) For DBN, it is not clear from the Materials and methods how the sequences are aligned since TCRs have variable length.

We have added an explanation of how the sequences are aligned.

8) To train a DBN, the authors construct the negative set from TCRs of healthy donors which were not in the expanded positive set. A natural negative set would be the patient-specific TCRs that were present in the pre-sensitized sample but did not expand. Authors should check how their results are robust to this choice of negative set.Also, TCRs are somewhat individual-specific and so using a negative set from different donors might be problematic and create an "easy" task for classification. This can be fixed by using donor specific training sets.

We have changed the control set to TCRs from the pre-sensitization repertoires of the same donors as the expanded set. As suggested by the reviewer, this is indeed a more difficult classification task, and we have found that the DBN is very good at telling individuals apart !!! With the new control set, classifying individual sequences is somewhat harder for the DBN, but it is still very successful at classifying sets of sequences. We have included both the results for sets and single sequences in the new Figure 7C.

What is the ratio of positive to negative set? Again, how robust is the performance of DBN to this training ratio. Authors should provide access to their training and test sets as a separate supplementary file.

The positive and negative sets are the same size, and running the model with shuffled class labels produces around 50% accuracy. We have added this to Materials and methods. The positive and negative sets are submitted as Supplementary files.

9) Can authors discuss in more detail the TCR features that are informative for DPC classification. Do the authors expect same classifier performance for other tasks (e.g. distinction of TCRs specific for certain viral epitope from TCRs specific for other epitopes), or high accuracy of DBN is limited to the special case of DPC-recognizing TCRs?

We are writing a manuscript fully focused on the DBN and we would prefer to leave the discussion of other data sets etc. to this other manuscript, as it is not really relevant to this manuscript.